# An efficient approach to identifying anti-government sentiment on Twitter during Michigan protests

Hieu Nguyen and Swapna Gokhale

Computer Science and Engineering, University of Connecticut, Storrs, Connecticut, United States

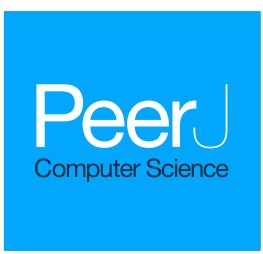

## ABSTRACT

Trust in the government is an important dimension of happiness according to the World Happiness Report (*Skelton, 2022*). Recently, social media platforms have been exploited to erode this trust by spreading hate-filled, violent, anti-government sentiment. This trend was amplified during the COVID-19 pandemic to protest the government-imposed, unpopular public health and safety measures to curb the spread of the coronavirus. Detection and demotion of anti-government rhetoric, especially during turbulent times such as the COVID-19 pandemic, can prevent the escalation of such sentiment into social unrest, physical violence, and turmoil. This article presents a classification framework to identify anti-government sentiment on Twitter during politically motivated, anti-lockdown protests that occurred in the capital of Michigan. From the tweets collected and labeled during the pair of protests, a rich set of features was computed from both structured and unstructured data. Employing feature engineering grounded in statistical, importance, and principal components analysis, subsets of these features are selected to train popular machine learning classifiers. The classifiers can efficiently detect tweets that promote an anti-government view with around 85% accuracy. With an F1-score of 0.82, the classifiers balance precision against recall, optimizing between false positives and false negatives. The classifiers thus demonstrate the feasibility of separating anti-government content from social media dialogue in a chaotic, emotionally charged real-life situation, and open opportunities for future research.

## INTRODUCTION & MOTIVATION

The World Happiness Report indicates that the extent to which people trust their governments and institutions plays an important role in their happiness and well-being (*Helliwell et al., 2021*). Communities which show high levels of trust are happier and more resilient in the face of a wide range of crises (*Skelton, 2022*). In the physical world, fringe groups that seek to question this trust by raising doubts in people's minds about the intent and motives of the government have always existed (*UN, 2012*). However, due to logistical reasons, these fringe groups in the offline era operated within a local scope, reaching only limited audiences. In the modern world, however, social media platforms have given easy, accessible, and approachable ways to these groups to spread their hateful, radical and

Corresponding author
Swapna Gokhale,
swapna.gokhale@uconn.edu

anti-government sentiment beyond the local boundaries of their operating areas to a far greater audience (*UN, 2012*).

The COVID-19 pandemic was one of the biggest health crises that we have seen in a century (*Skelton, 2022*). To control the spread of the virus and to keep people safe, state and local governments enacted many public health interventions such as masking, social distancing, and lockdowns. Of these, widespread lockdowns which ordered businesses and schools to shut down, and forced people to stay home were the most disruptive economically and socially. In the initial few weeks, many people viewed these orders as a necessary evil to protect our healthcare systems from being overwhelmed. As a result, they complied with these orders grudgingly, even though they viewed these draconian measures with skepticism.

A few weeks into the lockdown, however, anti-government sentiment fermented among far-right extremists. These extremist groups called for protests in many state capitals across the United States to oppose lockdown orders and to compel their governments to scale them back. These protests were considered ill-advised and unsafe by public health experts, and were covered extensively in national and international news.

Social media platforms offer a conduit for people to voice their opinions, thoughts and beliefs. Hence, these politically motivated and turbulent protests in the offline world also led to vigorous dialogue and exchange online as these platforms were the only channels of communication available to people to stave off their social isolation during the pandemic. Supporters of the protests often shared extreme and radical views, sowed distrust about governments' measures, contemplated ballot recalls to overthrow elected governments, and threatened violence against elected officials. Such anti-government rhetoric is often used for fundraising and recruitment purposes, and if left unchecked online, can lead to social unrest, violence and bloodshed. This was amply exemplified when subsequent protests involved guns, led to storming the Capitol building and eventually to a plot to kidnap the governor of Michigan (*Bogel-Burroughs, 2020*). However, if detected and mitigated earlier, such violent, out-of-control expression of anger and resentment may be prevented.

This article presents a classification framework to identify tweets which espouse extreme, anti-government views during politically charged protests in Lansing, Michigan. Tweets were collected during two separate anti-lockdown protests, and these tweets were then annotated as anti-government or non anti-government. A rich set of features was computed from both the structured and unstructured data collected along with these tweets. These features captured the content of the tweets, how Twitter users interacted with these tweets, and the properties of the tweets' authors. Three levels of feature engineering based on statistical significance, importance measures, and principal components was used to narrow down subsets of significant features that contribute meaningfully towards separating the tweets into anti-government and non anti-government groups. These subsets of features were used to train popular machine learning classifiers. The results showed that the classifiers could efficiently identify tweets that harbor anti-government sentiment with an accuracy of around 87%, with a training time of only a few seconds. The classifiers could also trade between precision and recall well with a F1-score of 0.82,

balancing between false positives and false negatives. These results are particularly noteworthy because unlike other studies where radical content is shared by a particular group which embraces a narrow philosophy (and hence most content either condemns or condones that philosophy), in this case anti-government tweets are shared within the broader context of COVID-19 measures, local and non-local politics, and the tactics employed during the protests. The results thus demonstrate that anti-government rhetoric can be separated from broad and general social media conversations, and opens opportunities for future research in this area.

The rest of the article is organized as follows: Section 2 compares related research. Section 3 summarizes the steps in data preparation. Section 4 presents computation of features. Section 5 provides an overview of classifiers and performance metrics. Section 6 discusses the results. Section 7 concludes the article and offers future directions.

## RELATED RESEARCH

Presently, social media platforms have been exploited to share and spread radical and extreme ideas that sow suspicion and hatred in order to destabilize democratically-elected institutions and governments. Alongside, research to detect such content from social media feeds has also gained traction to stem this rising tide. In this section, we compare and contrast contemporary efforts that have appeared in the literature on the topic of identifying radical content from social media dialogue.

*Miranda et al. (2020)* describe a technique based on support vector machines to detect radicalism in the content shared on Twitter in Indonesia. *Qi et al. (2019)* analyze Twitter and Reddit data around Hong Kong protests to identify influencers. *Wolfowicz et al. (2021)* differentiate between Facebook profiles of violent and non-violent radicals. *Ahmad et al. (2019)* present a deep learning-based sentiment analysis technique to classify extremist and non-extremist tweets. At the level of user accounts, *Abd-Elaal, Badr & Mahdi (2020)* present a classification framework to detect ISIS and non-ISIS accounts on Twitter. Another study related to ISIS is by *Mussiraliyeva et al. (2020)*, where they seek to detect ISIS-related language in Kazak using ensemble learners. *Yasin et al. (2021)* use unsupervised k-means clustering to group tweets into extremist and non-extremist content. *Araque & Iglesias (2020)* detect radical content from ISIS accounts by comparing it against the content from news outlets such as the New York Times and CNN. *Wu & Gerber (2018)* explore the predictive power of social media data in determining the onset of civil uprisings during the Egyptian revolution, whereas a recent study summarizes the literature on this topic (*Grill, 2021*).

Most of the above studies have been conducted on radical extremism beyond the U.S. shores and in regions where this prevalence is believed to be high. In recent years, however, social media platforms have been exploited within the United States to erode trust in political institutions (*Nguyen & Othmeni, 2021*) by spreading extreme, off-mainstream content. A social media platform that has gained notoriety for sharing and propagating such content under the guise of free speech is Parler (*Aliapoulios et al., 2021*). Supporters of radical groups such as Proud Boys, Boogaloo Bois, and QAnon have also been particularly active in certain regions of the country and on platforms such as Twitter and Reddit, and

their activities offline and online have been studied (*DeCook, 2018*; *Klein, 2019*; *Reid, Valasik & Bagavati, 2020*; *Fahim & Gokhale, 2021*). These works focus on radical and extremist content shared by groups that espouse a particular cause or a philosophy, and hence, most of the supporting chatter either promotes or criticizes that philosophy. For example, users of Parler are concerned about free speech, and Proud Boys is a far-right, neo-fascist, exclusively male organization.

In our work, however, anti-government discourse is embedded in a broader context, ranging from protesting COVID-19 public health measures to campaigning for (or against) local and non-local politicians and their governing philosophies, to criticizing (condoning) the tactics employed during the protests. We build a framework that can mine social media feeds to provide unique and early insights into people's radical opinions and thoughts, regardless of such context. This framework can help prevent violence and social unrest, especially in turbulent and chaotic circumstances such as those brought about by the COVID-19 pandemic, which was amply exploited by extremists to spread mistrust and hatred about the government and its policies (*Clarke, 2022*).

## DATA PREPARATION

This section discusses three steps in the preparation of data: collection, annotation, and pre-processing.

### Data collection

During the months of April and May 2020, anti-lockdown protests were organized in many states, including North Carolina, Michigan, Pennsylvania, Virginia, and California (*Andone, 2020*). Of these, the protests that occurred in Michigan gained ill reputation for many reasons. First, the crowds of protesters in other states were in the hundreds while the Michigan protests were the largest, attracting thousands of protesters. Second, while protesters in other states simply gathered on the streets, those in Michigan engaged in many violent and questionable tactics that threatened the health and well-being of many. Third, the protesters wanted to draw attention to the conflicting political situation in Michigan and its status as a battleground state in the 2020 presidential election. This conflict arose because Michigan was led by a Democratic governor but voted for President Trump in the prior (2016) presidential election. Thus, the protesters included members of Women for Trump, mainstream Republicans, anti-vaxx and gun rights advocates, Proud Boys, and Boogaloo Bois (*Ecarma, 2020*; *Wilson, 2020*). These factors drew media attention, followed by considerable volumes of conversations on social media platforms. Therefore, although protests were conducted in many states, we chose Michigan as a prominent example of anti-lockdown protests in the U.S. The specific circumstances surrounding the two protests, which were considered in building a coding guide for the annotation of tweets, were as follows:

- **Operation Gridlock:** This was the name given to the first protest. It was organized by a Facebook group with the same name, created by the Michigan Freedom Fund and Michigan Conservative Coalition. Close to 3,000 people showed up, the protest lasted

8 h, and the protesters blocked ambulances from reaching the only Level I trauma center at Sparrow Hospital. Most stayed in their cars, jammed the streets around the capitol building, and caused delays during a shift change at the hospital. About 150 protesters spilled on the lawn of the Capitol, flouting social distancing and masking guidelines. Protesters carried confederate, Nazi, and American flags (*Berg & Egan, 2020*).

- **Michigan Protest:** During the second Michigan protest, hundreds of protesters carried firearms, dressed in camouflage and military garb, gathered at the Capitol, and many managed to enter the building. Thus, the second protest took a more violent tone. It was organized by the conservative group American Patriot Council. Confederate flags, swastikas, and nooses were present at this protest too (*Mauger, 2020*).

We collected tweets a few days following the two Michigan protests on April 15, 2020, and April 30, 2020. Corresponding to the respective trending hashtags, we used *#operationgridlock* for the first sample and *#michiganprotests* for the second sample. The tweets were collected using the rtweet API (*Kearney et al., 2020*). Each time, the sample resulted in about 4,000 tweets.

## Data annotation

The objective of our research is to identify anti-government, deviant content from the rest of the dialogue because such content seeks to undermine the faith and trust in the government and its policies. This loss of trust may make the government's job of protecting the people considerably harder. Therefore, to study our research question, we chose to label the tweets into two groups, one group consisted of anti-government tweets, and the second group included tweets that are not against the government or non anti-government. We note that the second group may contain a mix of pro-government and neutral tweets, and if our research question were stance detection (*Cotfas et al., 2021*), we would further split the second group into these two categories. However, because the scope of our research question is limited to detecting tweets that sow resentment and suspicion against the government, we chose to combine the tweets that voiced support for the government along with the neutral tweets. Two additional reasons also motivated us to retain the pro-government and neutral tweets into a single class. First, generally, pro-government content may be considered suspicious and propagandist in autocratic or non-democratic regimes (*Stukal et al., 2022*; *Caldarelli et al., 2020*). However, governments in the U.S. at all levels (local, state, and federal) are elected democratically through free and fair elections. Thus, in this dialogue, pro-government content affirmed support for the public health restrictions that were implemented and was not viewed as propaganda. Second, our human coders found separating between these two types of tweets confusing, yielding a lower agreement between them. Therefore, we sought to annotate each tweet as either anti-government ('A') or non anti-government ('N').

To facilitate this annotation, we built a coding guide that the manual annotators could consult. This coding guide consisted of themes and the representative examples of both anti-government and non anti-government tweets that fitted each theme. Through an

extensive review of the news stories and opinion pieces, it was observed that most of the tweets fell along the following themes:

- **Tactics and Circumstances:** These tweets referred to the tactics employed by the protesters, and the other circumstances surrounding the protests. Although these tactics were disruptive and even violent; naturally, anti-government tweets praised them for the inconvenience they caused and the threatening/intimidation situations they produced. On the other hand, non anti-government tweets condemned them for the same reasons.
- **Local Politics:** These tweets mentioned local political figures in Michigan, with Governor Whitmer appearing predominantly. DeVos was another highly visible Republican family in Michigan with a significant presence and was believed to have sponsored the protests (*Hernandez, 2020*). Anti-government tweets denigrated the governor as a dictator and a Nazi, whereas non anti-government tweets stood with her in solidarity.
- **Non-local Politics:** These tweets cast the protests in Michigan as a part of the broader landscape and encouraged people in other states and nationally to engage in similar resistance and rallies to ease COVID-19 restrictions. Nationally visible Republican and Democratic leaders and governors of other states were mentioned in these tweets.
- **COVID-19:** These tweets explicitly referred to COVID-19. Anti-government tweets questioned the motive behind the public health measures and expressed skepticism about the seriousness of the virus. Non anti-government tweets mostly voiced concern about how these protests, which also came with rebelling against the public health guidelines such as social distancing and masks, would affect the trajectory of the number of cases.
- **Political Ideology:** These tweets were ideologically inspired; anti-government tweets praised the protesters as patriots and defenders of individual liberties and freedoms, while non anti-government tweets were critical of the protesters as white supremacists and racists.

Tables 1 and 2 show representative examples of anti-government and non anti-government tweets for each theme for Operational Gridlock and Michigan Protest data sets respectively. This coding guide was given to two annotators, who labeled each tweet as either anti-government ('A') or non anti-government ('N'). We eliminated duplicates before labeling. Both annotators had to agree upon the label for a tweet to be included in the final *corpus*. The disagreement between the two annotators eliminated approximately 450 tweets from each data set. The coders only coded each tweet as either anti-government or non anti-government; they did not identify the specific theme associated with the tweet. Therefore, although tweets from all the five themes were included in the analysis, it is not feasible to provide the split of the tweets into these five themes. The collective summary of the tweets from all five themes and their distribution between anti-government and non anti-government groups in the individual and the combined data set is summarized in Table 3.

**Table 1  Themes & example tweets—Operation Gridlock.**

**Theme I: Tactics & circumstances**

A   This doesn't even begin to show the number of people in Lansing. Block and blocks of people siting in there cars. It didn't look like so many by the capital bc some streets were clo we d off. Vast majority stayed in their cars.

N   A friend took this from a hospital in Lansing, Michigan. Apparently the protesters blocked an ambulance from getting to the hospital. https://t.co/RxpA9S4TvL

**Theme II: Non-local politics**

A   @GovInslee Jaydid you pay attention to Lansing tonight?

N   Yet these #Trump supporters call themselves #ProLife—blocking Hospital Workers from getting to work. #Lansing https://t.co/Tl42VMk22E

**Theme III: Local politics**

A   I'm thinking Lansing MI doesn't like their gov and her non-essential bullshit???

N   @GovWhitmer I stand with Governor Whitmer. It's nice to have a Governor actually care about the people of her state!!! I'm embarrassed by the selfish assholes protesting in Lansing today.

**Theme IV: COVID-19**

A   TRUTH IN NUMBERS. FLU BEING REPLACED FOR COVID $$$ LIVE SHUTDOWN PROTESTS IN LANSING MICHIGAN #COVID1984 #EndTheLockdown #F…

N   Presumably Lansing will now be Michigan's next virus hot spot.

**Theme V: Political ideology**

A   #MichiganProtest is not about going out to eat or getting a haircut—its about govt restricting our #NaturalRights to #Freedom #Liberty. @NatlGovsAssoc #gretchenwhitmerisa #Fascist #oppressor @GovWhitmer #ProtestLockdown #ProtestTyranny #WeveHadEnough

N   If this virus has taught me anything its that Americans literally have zero fucking clue what the constitution actually says. #COVID #MichiganProtest

# FEATURE COMPUTATION

The Twitter API returns both structured and unstructured data that represents the properties of the tweets and their authors in addition to the text, which is, of course, the core content of the tweet. Figure 1 shows the high-level processing pipeline of the steps involved in taking this raw data and converting it to features. Broadly speaking, we have three types of data, the text of the tweets, the parameters representing how Twitter users interacted with these tweets, and the inherent characteristics and activity level of the authors. In the next subsections, we elaborate on how each of these different types of data were mapped to features using Fig. 1 as the guide. For the numerical features, statistical significance between the two groups is assessed using the two-sample t-test (*Zafarani, Abbasi & Liu, 2014*).

## Text features

The text of the tweets is the most important piece of the data. Therefore, it is no surprise that the text's content, its syntactical presentation, and the hidden underlying emotion all contribute to determining whether a tweet is anti-government or otherwise.

The content of the tweet conveys the actual message of the author. This message is revealed only after the noise is separated from the tweet, which was achieved through the following pre-processing steps. First, we removed the uninformative symbols such as mentions, hashtags, and hyperlinks. After this step, words that comprised the hashtag and mentions were regarded as a part of the tweet's text. Then, we eliminated stop words that

**Table 2 Themes & example tweets—Michigan protest.**

**Theme I: Tactics & circumstances**

A #Patriots, please use some restraint.. Whitmer is trying to goad Patriots into violence so she can justify #gun grab. #MichiganProtest #2A #OathKeepers @TheJusticeDept

N My better judgement tells me that if you show up at a State Capital building decked out in riot gear and an AR 15, you would be immediately arrested, locked up and charged with assault? What happened to those days?? #MichiganProtest https://t.co/slDvRVNYod

**Theme II: Non-local politics**

A Leave it a vile #Democrat to equate protesting to racism. #LiberalismIsAMentalDisease #OPENAMERICANOW #MichiganProtest https://t.co/GSPqpIsQ7n

N Why does @realDonaldTrump have a habit of calling people with confederate flags, swastikas, and nooses "Very fine people" or "very good people." Hmm. #Charlottesville #MichiganProtest.

**Theme III: Local politics**

A Fuck off @GovWhitmer, you're trash and so are your politics. #MichiganProtest #BidenTheRapist #MeTooUnlessItsBiden #MeToo https://t.co/EL7weMR9Fb

N Flint hasn't had drinking water for a decade but don't make #Michiganers stay home for a month! #MichiganProtest #worthless #loserswithguns

**Theme IV: COVID-19**

A @GovWhitmer awful response to COVID will dwarf her confusion to her self-induced Michigan SHUTDOWN RECESSION plan—a massive train wreck. #COVID19 #MichiganProtest

N @VP was reportedly told by hospital officials that a mask was a requirement and he still refused to wear one, while visiting #COVID_19 patients. To feed into the redoric of violant #DomesticTerrorists that participated in last nights #MichiganProtest for

**Theme V: Political ideology**

A All I can say is God Bless those patriots who grabbed their AK-47, filled up their Dodge Ram with their stimulus check and headed to Lansing to ignore the CDCs social distancing guidelines! These are the REAL heroes!

N But we have dummies in Lansing marching for white supremacy…smh

**Table 3 Summary of data sets.**

| Protest | Total | Anti-government | Not Anti-government |
|---|---|---|---|
| Operation Gridlock | 3,570 | 931 (26.08%) | 2,639 (73.90%) |
| Michigan protest | 3,596 | 956 (26.59%) | 2,990 (83.15%) |
| Total | 7,166 | 1,887 (26.33%) | 5,629 (78.55%) |

commonly occur in the English language. These are the words that appear in NLTK's English stop word dictionary. We also added words such as "u" and "ur"; these words are commonly found in tweets but are not in the English dictionary. Added to this list of stop words were context-specific words such as "lansing", "michigan", "people" and "today", which occurred with similar frequency in both groups and hence, were likely to not contribute to the classification. Finally, the words were stemmed down to their stems using the PorterStemmer in the NLTK library (*Willett, 2006*). Stemming involved truncating the words down to their roots, for example, "writing", "writes", "written" got truncated to "write". The text remaining after pre-processing was then split into a list of words.

Figure 2 shows the word cloud visualizations built from the topmost unique words (remaining after pre-processing) in anti-government and non anti-government tweets. Anti-government tweets promote the ideological perspective that lockdowns are an

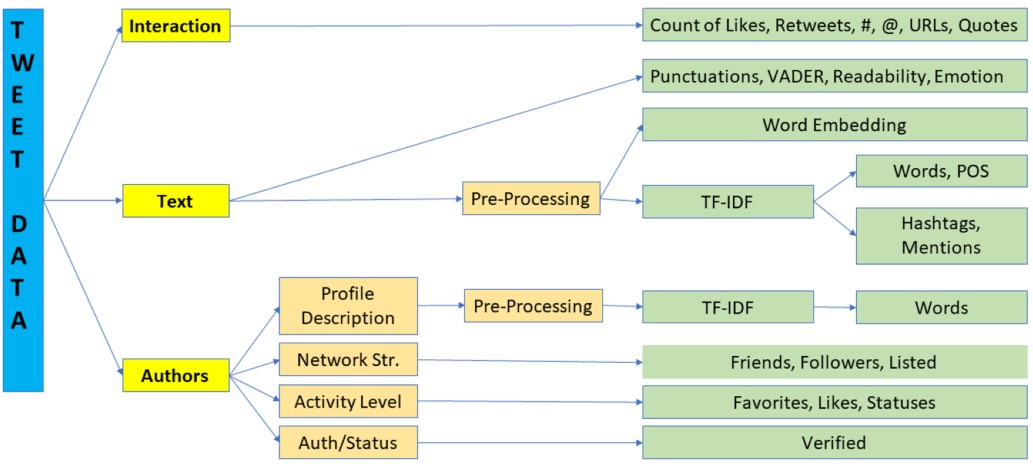

**Figure 1** Feature computation processing pipeline.

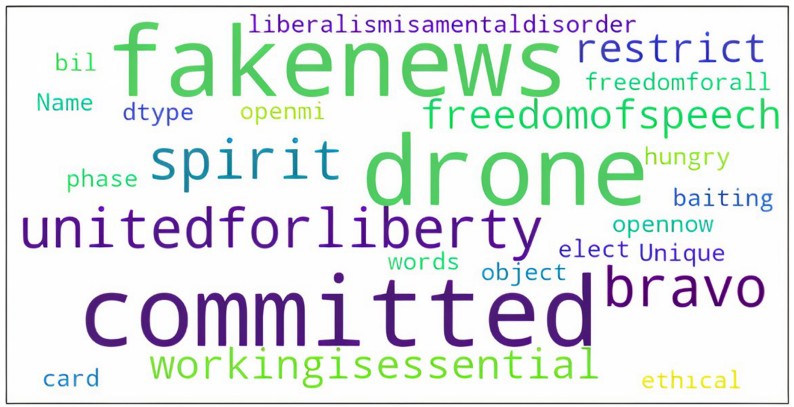

A

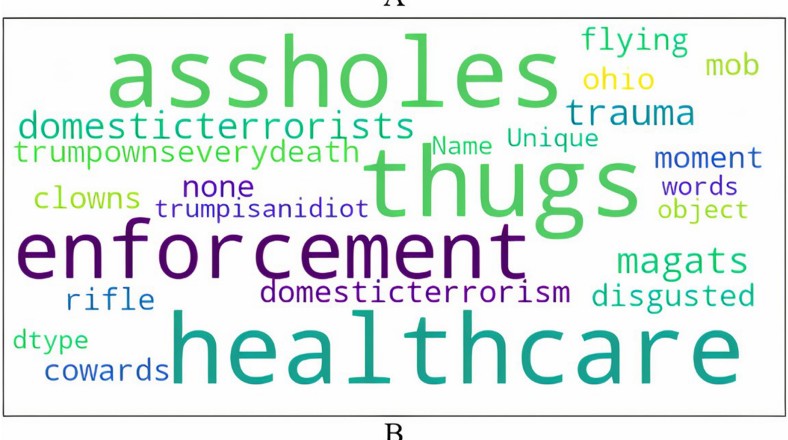

B

**Figure 2** Unique words in tweets.

infringement on individuals' liberty and freedom. They mock liberalism, call for an end to the lockdowns by claiming that working is an important economic activity (all non-essential businesses were ordered to be shut down), and embrace skepticism about the virus. On the other hand, tweets that are non anti-government insult the protesters by

referring to them as "assholes", "thugs", "clowns", "magats", "cowards", "mob", and "domestic terrorists". Some tweets also associate the protesters with President Trump and his political views as suggested by these examples. In the tweet *I live in Lansing…why I've seen is armed protesters…many of whom are KKK, Michigan militia and assorted assholes*, the term "assholes" is used in the context of Michigan militia that aligned with the president. Another tweet *These #maga thugs are blocking the entrance to Sparrow Hospital in Lansing, MI to protest the stay at home order, calling it Operation Gridlock* refers to the protesters as MAGA thugs. MAGA (Make America Great Again) was President Trump's election slogan. The president is also blamed for COVID deaths as seen by "trumpownseverydeath". There are many tweets that lament over the large number of deaths due to COVID, further alluding to the overall apathy and indifference of the president and his supporters to this issue; for example *Only 153 deaths today? Sounds like a great day for a Trump rally in Lansing. Sorry idiots Trump isn't there today. #OperationGridlock*. Disgust and disbelief in these ill-informed protests asserting freedom is also expressed.

We used *n*-grams/TF-IDF (Term Frequency Inverse Document Frequency) and word embeddings to map these unique words to features. In the *n*-grams method, a text sample is represented by the most frequent instances of every unique *n* continuous words as a dimension. We calculated bigrams from the pre-processed text, and the weight for each bigram was its TF-IDF score (*Zafarani, Abbasi & Liu, 2014*). Words that make up hashtags and mentions were also included in the computation of bigrams, and hence, TF-IDF scores. TF-IDF scores assign a higher weight to those bigrams that are the most important differentiators between anti-government and non anti-government tweets. We calculated TF-IDF vectors from our *corpus* for top the 2,000 relevant bigrams.

TF-IDF provides a weighted score based on statistical importance. However, it does not preserve the contextual relationship among the words. To represent the context between the words, we computed Word2Vec embeddings based on neural networks that map semantically related words to low-dimensional, non-sparse vectors (*Mikolov et al., 2013*). Using Gensim library (*Rehurek & Sojka, 2010*), our model mapped each word to a 10-dimensional vector, with a minimum count of 1, and the number of partitions during testing set to 8. Because our data size was small, we used the skip-gram model (*Nicholson, 2019*).

The syntax and other arrangements of the words in the tweets capture how the content is conveyed. Authors may use various punctuations such as question marks, exclamations, quotes, etc., and other markers such as emoticons and upper case letters to emphasize how strongly they feel about their content (*Araque & Iglesias, 2020*). They may also use higher proportions of certain parts-of-speech such as adverbs and verbs to express their passion (*Xu, 2014*). In face-to-face communication, facial expressions and body language usually provide additional clues about the underlying emotion and passion of the speakers and their intensity. These clues, however, are not present in written communications, including social media texts. Therefore, in written texts, these syntactical patterns and organizations of words can substitute for facial gestures and non-verbal clues. Moreover, prior research indicates that these non-textual parameters may differ between polarising and regular

tweets (*Araque & Iglesias, 2020*). To represent these features, we included counts of question marks, exclamation marks, periods, quotation marks, links, and capital words. We also computed TF-IDF scores for different parts of-speech (*Xu, 2014*), using the NLTK library (*Loper & Bird, 2002*).

We computed two readability scores, representing the ease with which readers can understand the tweets. Readability is determined by how complex the vocabulary is, its syntax, and how the content is organized into sentences and paragraphs (*Benoit et al., 2018*). These are the Flesch Reading Ease and Flesch-Kincaid Grade Level indices (*Talburt, 1985*), and their respective expressions are:

$$\text{Flesch Reading Ease Index} = 206.835 - 1.1015\left(\frac{W}{S}\right) - 84.6\left(\frac{L}{W}\right) \tag{1}$$

$$\text{Flesch Grade Level Index} = 0.39\left(\frac{W}{S}\right) + 118.0\left(\frac{L}{W}\right) - 15.59 \tag{2}$$

In these indices, $S$ represents the total number of sentences, $W$ represents the total number of words, and $L$ represents the total number of syllables in the text for which the indices are to be computed. $W$ captures the sentence length, and $L$ captures the word length. The philosophy behind using sentence length and word length is that longer sentences and words are more difficult to read and understand than shorter sentences and words.

Each index weighs these two factors differently, because of the different outputs they produce. The Flesch Reading Ease index produces a score between 0 and 100, which is then interpreted in 10-point intervals to determine readability. The higher the score, the easier it is to read the text, with scores between 100.0 and 90.0 suitable for a 5th grader, whereas scores between 10.0 and 0.0 suitable for a professional. On the other hand, the Flesch Grade Level index directly presents the score in terms of suitability for U.S. grade level (*Talburt, 1985*). Table 4 lists the values of these readability indices for anti-government and non anti-government tweets. As indicated by the *p*-values, there was a statistically significant difference in the Flesch Reading Ease index between the two groups, but the difference in the Flesch-Kinacid Grade Level index was statistically insignificant.

Tweets with disruptive information generally exhibit less emotion and sentiment and tend to be overall negative. By contrast, regular tweets that voice support for the democratic institutions may have a positive outlook and sentiment (*Araque & Iglesias, 2020*). These differences were quantified using the VADER sentiment scores computed from the text of the tweets (*Hutto & Gilbert, 2014*). The positive, negative, neutral, and compound scores for the tweets from the two classes are listed in Table 4. The difference in all the scores was statistically significant between the two groups. We also used scores for six emotions; namely, sadness, joy, love, fear, anger, and surprise computed using a pre-trained DistilBERT model. This model is a fast, cheap light transformer model based on the BERT architecture. The model is trained on the Twitter sentiment analysis data set and shows an accuracy of 93.8% and F1-score of 93.79% (*Saravia et al., 2018*). Table 4 shows that scores for the emotions of sadness, joy, and anger were statistically significant, whereas

**Table 4  Readability, sentiment & emotion features.**

**Readability**

| Parameter | A | N | *p*-value |
| --- | --- | --- | --- |
| Flesh reading ease | 68.625 | 72.916 | 0.0000 |
| Flesch-kincaid grade level | 9.283 | 9.100 | 0.1845 |

**VADER sentiment**

| Parameter | A | N | *p*-value |
| --- | --- | --- | --- |
| Vader negative | 0.157 | 0.212 | 0.0000 |
| Vader positive | 0.088 | 0.071 | 0.0000 |
| Vader neutral | 0.754 | 0.717 | 0.0000 |
| Vader compound | −0.156 | −0.353 | 0.0000 |

**Emotions**

| Parameter | A | N | *p*-value |
| --- | --- | --- | --- |
| Sadness | 0.046 | 0.062 | 0.0008 |
| Joy | 0.176 | 0.134 | 0.0000 |
| Love | 0.003 | 0.003 | 0.9140 |
| Anger | 0.737 | 0.764 | 0.0046 |
| Fear | 0.035 | 0.034 | 0.8253 |
| Surprise | 0.002 | 0.002 | 0.9252 |

scores for the emotions of love, fear and surprise were not significant between the two groups.

As shown in Fig. 1; sentiment, readability, and emotion scores were computed from the raw text prior to pre-processing, which is why they are all listed together in Table 4.

## Interaction features

When users tweet their thoughts, they expect other users to engage with and react to their tweets. Twitter users interact with the tweets in two public ways, liking (favoriting) and retweeting. Both likes and retweets may be viewed as forms of endorsement. Likes may be considered as a more tacit, passive method where the message contained in a tweet resonates with the user. Tweets liked by users are visible to their friends. On the other hand, retweeting is an active approach where users re-broadcast the tweets that their friends share to their entire network. Retweeting could be used to show the intention of listening and agreeing with the tweet owner's point of view (*Tweettabs, 2022*; *Hajibagheri & Sukhthankar, 2014*). Both these forms of interactions boost the diffusion and spread of the tweets, and therefore we use these parameters to measure the degree of interactions with a tweet. The table also compares these parameters for quoted tweets. Table 5 shows that average number of likes and retweets is significantly higher for anti-government compared to non anti-government tweets. The same trend holds for quoted tweets, and these differences are significant at the 5% level.

Users may also deliberately engineer their tweets to improve interaction (*Oehmichen et al., 2019*). We discuss the meaning of these actions and whether there exist any

**Table 5 Interaction features.**

| Parameter | A | N | *p*-value |
|---|---|---|---|
| #likes | 81.37 | 12.98 | 0.0004 |
| #retweets | 23.82 | 5.69 | 0.0017 |
| #likes (Q) | 21,918.82 | 7,729 | 0.0000 |
| #retweets (Q) | 6,385.70 | 2,900.70 | 0.0000 |
| #hashtags | 1.13 | 1.24 | 0.0210 |
| #mentions | 0.49 | 0.78 | 0.0000 |
| #quotes | 0.13 | 0.19 | 0.0158 |

quantifiable differences in these acts between the two groups. First, users may annotate tweets with one or more hashtags; adding a hashtag to a tweet allows other users to find their tweets and to interact with them. Adding a hashtag also builds a community of users discussing the same topics. Twitter also uses hashtags to calculate trending topics of the day, which further encourages users to post and join these communities (*Hajibagheri & Sukhthankar, 2014*). In the table, we show the average number of hashtags per tweet for both groups. Users may also mention other users; this can be viewed as a social activity, where one user is taking account of other user(s), and is oriented towards the course. Mentions can also be used to improve visibility. Finally, the tweets may also quote other tweets with additional comments of their own. These comments may either support the content of the original tweet or refute it. Either way, it can enhance the visibility of their own tweet, especially if the quoted tweet is from a prominent or a verified account (*Park, Compton & Lu, 2015*). Finally, authors may also craft URLs into their tweets to support their point of view with scientific or literary evidence or news articles. Although the magnitude of the difference in the numbers of hashtags, mentions, and quotes did not appear too large between the two groups, the difference was still significant at the level of 5%.

## Authors' features

Data regarding the authors of the tweets is of two types; their profile descriptions and how they behave over the platform in terms of sharing their own thoughts as well as responding to others' content. We extracted features from both these types of data because prior research shows that both these factors, namely, whom the authors claim to be and how they act, will influence how far their tweets will spread (*Mann, Gaines & Gokhale, 2022*; *Oehmichen et al., 2019*).

The profiles of the authors were pre-processed through the same pipeline used for the text of the tweets. Word cloud visualizations were built from the resulting text, and TF-IDF features representing these profiles were extracted. These word clouds for authors who tweet anti-government and non anti-government content are shown in Fig. 3. The few words that stand out from the profiles of authors who shared anti-government tweets include *conserv*, *american presid*, *god*, *maga* and *family*. These words indicate that authors with conservative ideology, harboring support for the constitution, faith and family shared

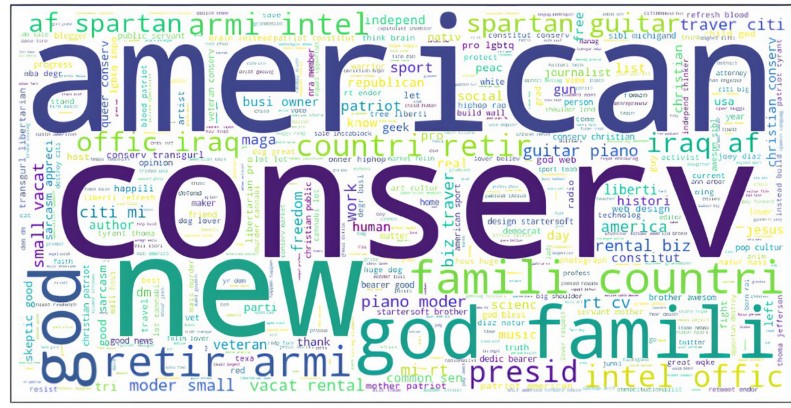

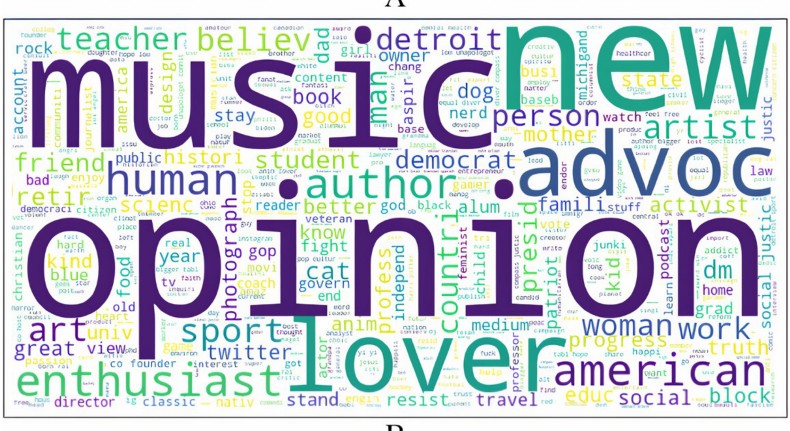

**Figure 3 Author profiles of tweets.**

anti-government tweets. They may also be ardent supporters of President Trump and his populist campaign theme MAGA. On the other hand, prominent words in the word cloud of non anti-government tweets' authors are *human*, *social*, *democrat*, *artist*, *teacher*, *author*, *advocate*. These words point to those whose philosophy is liberal-leaning, may be employed in service-oriented professions, and are advocates of social justice and causes.

Next, we divided the parameters reflecting authors' behavior into those that represent the strength of their network, their authenticity and status, and their level of activity. Wherever available, parameters in these three groups were also compared for authors of quoted tweets. These three groups of parameters are discussed below (in this discussion original tweet refers to the tweet sampled by the API that appears in the *corpus*), and their average values for both anti-government and non anti-government tweets are in Table 6:

- **Network strength:** The strength of the authors' network can be assessed by the numbers of friends and followers. Generally, tweets of those authors who have larger networks of friends and followers can expect greater interaction and visibility. The numbers of friends and followers are compared both for authors of original and quoted tweets. The listed count indicates the number of other users who have added an author to their list and can be an indicator of popularity (*Tweettabs, 2022*). It is thus reasonable to believe

**Table 6 Authors' features.**

**Network strength**

| Parameter | A | N | *p*-value |
|---|---|---|---|
| #Friends | 2,373.93 | 3,338.45 | 0.0014 |
| #Followers | 6,854.95 | 6,115.54 | 0.7929 |
| #Friends (Q) | 19,146.97 | 23,073.83 | 0.6044 |
| #Followers (Q) | 413,655.74 | 1,766,298.38 | 0.0520 |
| #Lists | 58.96 | 46.37 | 0.2013 |
| **Activity level** | | | |
| Parameter | A | N | *p*-value |
| #statuses | 26,542.22 | 31,987.65 | 0.0480 |
| #favorites | 27,609.38 | 25,473.09 | 0.2678 |
| #statuses (Q) | 42,560.14 | 51,456.97 | 0.2129 |
| **Authenticity/Trust** | | | |
| Parameter | A | N | *p*-value |
| % Is verified | 0.0135 | 0.0257 | 0.005 |

that tweets of authors with greater listed counts will be more popular and will receive more likes and retweets. The table indicates that only two parameters, namely, the number of friends of the authors of original tweets and the number of followers of the authors of quoted tweets, are significant. The difference in the other parameters is insignificant between the two groups.

- **Activity level:** One of the main indicators of the degree to which the authors are active on the platform is the number of status updates they have shared through the entire period that their accounts have been active. Status updates were compared for the authors of both the original and quoted tweets. Authors of non anti-government tweets have posted a significantly greater number of status updates compared to the authors of anti-government tweets. However, this difference is insignificant for authors of quoted tweets from both groups. Other secondary indicators include the number of times they have liked tweets from their friends and followers. Authors who prolifically react to tweets that appear on their feeds are likely to invite similar altruistically reciprocal relationships from their friends and followers (*Oehmichen et al., 2019*). Thus, the number of likes a tweet receives may have a high positive correlation with the number of tweets the author may have liked. However, there is no significant difference in the number of likes (listed as the number of favorites in Table 6 according to the nomenclature used by the Twitter API) by the authors of tweets from both groups.

- **Authenticity/Trust:** Tweets from high-profile, celebrated authors may attract a lot more attention, probably because the general public implicitly believes that the content shared from their accounts is more trustworthy and authentic. Moreover, these authors tend to have much larger networks of followers than those who are not celebrities. These popular authors who enjoy celebrity status tend to have accounts verified by Twitter; and hence, whether a tweet is shared from a verified account can be a factor in

influencing its spread. Thus, the table also compares the percentage of tweets shared from verified accounts for both classes. In the absolute sense, the percentage shared from verified accounts is trivial for both anti-government and non anti-government tweets. However, the difference between the percentage is statistically significant, as indicated by the $p$-value.

## CLASSIFIERS AND PERFORMANCE

We chose the following popular models for classification. Multiple models are used since each model uses a different philosophy to arrive at a decision, and it is impossible to tell *a priori* which model will offer the best performance for a specific data set. The collection of models included an ensemble learner (Random Forest), a simple, basic learner (Logistic Regression), a simple, sophisticated learner (Support Vector Machine), a neural network (Multi-Layer Perceptron), and a pre-trained transformer model (DistillBERT). Model implementations in the Scikit package were used (*Pedregosa et al., 2011*), and their hyperparameters are listed below:

- **Random Forests (RF):** Random Forests is an ensemble learning method, where the underlying weak learner is a Decision Tree (*Liaw & Wiener, 2002*). It uses bagging to reduce variance by generating a number of decision trees with different training sets and parameters. The parameters of the model are as follows. Each forest consisted of 100 trees, the maximum number of features used to grow each tree in the forest is set to the square root of the total number of features (approximately 25–30 when all the features are employed), and each decision tree is not pruned.

- **Support Vector Machines (SVM):** Support Vector Machines (SVM) is a powerful classification technique that estimates the boundary (called hyper-plane) with the maximum margin (*Suykens, Lukas & Vandewalle, 2000*). We used SVMs with both linear and RBF kernels and L2 regularization. The regularization parameter $C$ was set to 1, and kernel coefficient $\gamma$ was set to scale.

- **Logistic Regression (LR):** One of the basic and popular algorithms to solve classification problems, this is named as such because of the Logit function that forms its basis. The parameters are penalty–L1, tolerance for the stopping criteria–0.0001, the inverse of the regularization strength $C$–1.00 and the maximum number of iterations–100 (*Pedregosa et al., 2011*).

- **Multi-Layer Perceptron (MLP):** Multi-Layer Perceptron is a feed-forward Artificial Neural Network (ANN) that consisted of input, hidden, and output layers (*Delashmit & Manry, 2005*), set to 10, 8, 5 and 2 respectively. We used rectifier linear unit (ReLu) instead of the sigmoid activation function to handle the problem that the derivative of the activation function rapidly approaches zero. This problem with the derivative is common in deep neural networks.

- **DistilBERT (D-BERT):** BERT (Bidirectional Encoder Representation from Transformers) is a deep learning model in which all outputs are connected with each input, and the weightings between them are dynamically calculated in the attention

layers (*Devlin et al., 2018*). This characteristic allows the model to understand the context of the words based on their surrounding words as compared to directional NLP models. We employed DistilBERT, a compact version of BERT where the model has 40% fewer parameters than BERT while preserving over 95% of BERT's performance (*Sanh et al., 2019*). The parameters of the DistilBERT model include: vocabulary size (30,522), max position embeddings (6), number of layers (6), number of heads (12), dimensions (768), number of hidden dimensions (3,072), dropout (0.1), attention drop out (0.1) and activation function (gelu) (*Sanh et al., 2019*).

Our main objective was to identify anti-government tweets; hence, to define the performance metrics, we designated the anti-government and non anti-government classes as positive and negative, respectively. Tweets could thus be classified into four groups–true positive (TP) (anti-government labeled as anti-government), true negative (TN) (non anti-government labeled as non anti-government), false positive (FP) (not anti-government labeled as anti-government), and false negative (FN) (anti-government labeled as not anti-government). These four groups led to the following metrics to compare classifier performance:

- **Accuracy (A):** Accuracy was defined as the percentage of tweets that are labeled correctly.

$$\text{Accuracy} = \frac{TP + TN}{TP + FP + TN + FN} \tag{3}$$

- **Precision (P):** Precision measured the percentage of tweets that were actually anti-government out of all the tweets that were predicted as anti-government.

$$\text{Precision} = \frac{TP}{TP + FP} \tag{4}$$

- **Recall (R):** Recall measured how many of the anti-government tweets were actually labeled as anti-government.

$$\text{Recall} = \frac{TP}{TP + FN} \tag{5}$$

- **F1-score (F1):** F1-score balanced between Precision and Recall.

$$\text{F1} = 2 \times \frac{\text{Precision} \times \text{Recall}}{\text{Precision} + \text{Recall}} \tag{6}$$

Precision is the percentage relevant from the set detected and recall is the percent relevant from within the global population (*Zafarani, Abbasi & Liu, 2014*). Precision is important when the cost of a false positive is high. Applying symmetrical logic, recall would be important when the cost of a false negative is high. When identifying tweets with anti-government sentiment, a false positive implies that a non anti-government tweet is labeled as anti-government, whereas a false negative implies that an anti-government tweet is labeled as non anti-government. In false positive labeling, because a non anti-government tweet may be labeled as anti-government, it may be subject to one or more stringent misinformation policies such as being tagged with a warning label or turning off the likes and retweets to curb their spread, or ultimately, demoting or removing the tweet altogether (*Twitter, 2021*). These measures may raise allegations of freedom of speech violations (*Parler, 2022*). On the other hand, false negative labeling implies that an anti-government tweet will be labeled as non anti-government. This tweet may propagate across the network unhindered, spreading anti-government agenda. However, it will steer clear of freedom of expression violations. Absent a clear threat, where there may be a compelling reason, such as an explosive political environment, to curb the spread of anti-government tweets, a balance may be sought between precision and recall to trade-off between the diffusion of anti-government sentiment and violating freedom of expression. F1-score provides this balance between the two metrics.

## RESULTS AND DISCUSSION

We split the entire *corpus* using stratified sampling into two partitions; training and test consisting of 80% and 20% of the tweets, respectively. Stratified sampling preserves the ratio of anti-government to non anti-government tweets in each partition. With each split, we conducted extensive experimentation with three levels of feature engineering. Each level of feature engineering draws upon the results from the previous level, and is designed to allow for an increasing selectivity of features.

The performance of the classifiers for the three experiments is summarized in Table 7. The table also includes the time taken to train each model. The first experiment included all features, except for those interaction and authors' features which were not statistically significant. Collectively, this set consisted of 8,203 features, and they identified anti-government tweets with an accuracy of 86% and F1-score of 0.81. All the models, except for decision trees, offered competitive performance. It was not surprising that the performance of decision trees was significantly lower because these simple learners are prone to over-fitting (*Liaw & Wiener, 2002*). It was perhaps more surprising that the simple logistic regression model came close to complex models such as the multi-layer perceptron and support vector machines. The superior performance of the logistic regression model could be because much of the classification decision was based on the textual content of the tweets, similar to the detection of hate speech (*Khan et al., 2021*), a conjecture that we confirmed through importance analysis.

Figure 4 shows the importance scores of the various groups of features computed using the random forest model. Guided by the feature map in Fig. 1, we further grouped features into coarser categories. In Fig. 4, social features include those extracted from the structured

**Table 7 Performance metrics.**

| Expt. | Model | Accuracy | Precision | Recall | F1-score | Training time |
|---|---|---|---|---|---|---|
| I | Linear SVC | 0.82 | 0.76 | 0.77 | 0.76 | 3 min 5 s |
| | Support vector | 0.85 | 0.75 | 0.82 | 0.77 | 10 min 44 s |
| | Decision tree | 0.72 | 0.64 | 0.64 | 0.64 | 9.43 s |
| | Random forest | 0.84 | 0.69 | 0.85 | 0.73 | 19.3 s |
| | Logistic regression | 0.84 | 0.78 | 0.78 | 0.78 | 10.7 s |
| | Multi-layer perceptron | 0.86 | 0.80 | 0.82 | 0.81 | 2 min 14 s |
| | DistilBERT | 0.84 | 0.82 | 0.76 | 0.78 | 1 min 43 s |
| II | Linear SVC | 0.83 | 0.79 | 0.77 | 0.78 | 1 min 43 s |
| | Support vector | 0.85 | 0.80 | 0.80 | 0.80 | 13.4 s |
| | Decision tree | 0.71 | 0.62 | 0.61 | 0.62 | 1.52 s |
| | Random forest | 0.83 | 0.69 | 0.84 | 0.72 | 5.47 s |
| | Logistic regression | 0.81 | 0.79 | 0.75 | 0.77 | 0.53 s |
| | Multi-layer perceptron | 0.85 | 0.78 | 0.8 | 0.79 | 15.8 s |
| III | Linear SVC | 0.83 | 0.80 | 0.77 | 0.78 | 6.01 s |
| | Support vector | 0.87 | 0.82 | 0.83 | 0.82 | 10.5 s |
| | Decision tree | 0.74 | 0.64 | 0.65 | 0.65 | 4.2 s |
| | Random forest | 0.82 | 0.67 | 0.83 | 0.70 | 16.8 s |
| | Logistic regression | 0.82 | 0.80 | 0.76 | 0.77 | 0.33 s |
| | Multi-layer perceptron | 0.86 | 0.78 | 0.82 | 0.80 | 8.79 s |

data related to the tweets and their authors. These comprise of the interaction parameters of the tweets, and the network strength, activity level, and the authenticity status of the authors. Auxiliary features consists of punctuation counts, VADER sentiment, emotion, and readability scores. The figure shows that TF-IDF scores extracted from the text of the tweets and the profile descriptions of the authors contribute about 74% to the classification. It was thus possible to hypothesize that only a small subset of the features would be sufficient to achieve the same classification accuracy. Therefore, in the second experiment, we selected the top 300 features and re-trained and re-evaluated the classifiers. The metrics for the different classifiers after employing feature selection indicate that this step does not improve the performance of the classifiers. However, the table shows an appreciable reduction in the training time by employing feature selection. In fact, the training time of the support vector classifier, which is the model that offers the best performance, reduced from 10 min 44 s when the entire collection of features is used to merely 13.4 s when only the top 300 features were selected.

In the third experiment, we reduced the dimensionality of the top 300 features chosen in the second experiment *via* principal components analysis (*Jolliffe & Cadima, 2016*). The performance of support vector classifier with RBF kernel increases by 1%, and the time to train drops slightly to 10.5 s. Thus, feature selection and dimensionality reduction together offer a distinct advantage of improving the efficiency of the classification by reducing the number of features without sacrificing performance. Moreover, precision and recall

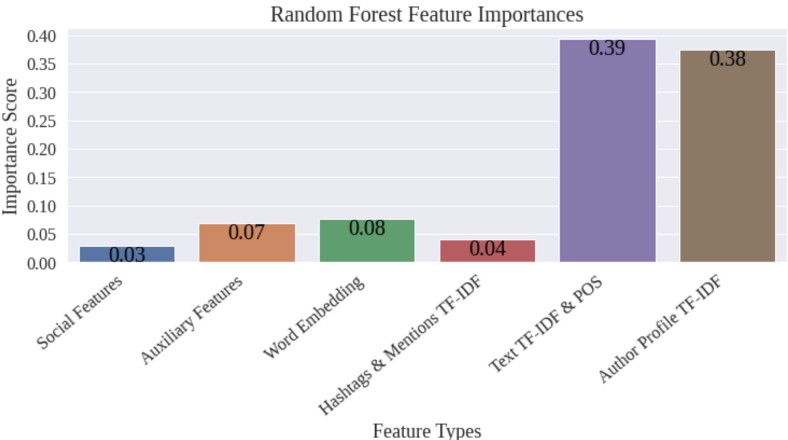

**Figure 4 Feature selection—importance scores.**

metrics are higher and more balanced, leading to a better F1-score when feature processing is employed than when it is not.

The transformer model DistilBERT is trained using the text from the tweets and authors' profiles, along with hashtags. Interaction and authors' features are not included in training the transformer model. Moreover, feature engineering cannot be applied to DistilBERT, and hence, the results of DistilBERT are compared with the performance metrics of the models from the first experiment in the table. The table shows that even when feature engineering is not employed for conventional machine learning models, DistilBERT is not the best performing model. In fact, it is outperformed by multi-layer perceptron. With subsequent feature engineering, DistilBERT is outperformed by both support vector and multi-layer perceptron classifiers. Thus, although pre-trained transformer models represent the state-of-the-art in natural language processing (*Nagda et al., 2020*), extensively trained conventional machine learning models with additional features extracted from meta data and careful feature selection and dimensionality reduction can outperform these models.

Our objective in this article was to build a classification framework that can detect deviant content that promotes an anti-government perspective. The basis of this framework is a set of features extracted from the structured and unstructured tweet data that we expect to remain invariant in conversations on various controversial topics and issues. Using these features, accompanied by feature selection and processing along with tuning of hyperparameters of machine learning models, we have also successfully applied this framework to detect tweets that spread anti-mask (*Cerbin et al., 2021*) and anti-vaccination (*Paul & Gokhale, 2020*), and those that support Proud Boys, an extremist, radical group (*Fahim & Gokhale, 2021*).

## CONCLUSIONS AND FUTURE RESEARCH

Radical extremists use social media platforms to spread their ideology effectively in order to gather a critical mass of followers to organize and execute violent and disruptive activities in physical spaces. Such anti-government sentiment can simmer on social media

platforms for a while, and can be a harbinger of disruption, bloodshed, and unrest downstream. Analyzing social media dialogue can detect these latent views and stop them for escalating further. This article presented a classification framework that detects anti-government sentiment in social media dialogue following the anti-lockdown protests in Lansing, Michigan during the COVID-19 pandemic. Using the tweets collected and labeled from two separate protests, we computed a rich set of features from both structured and unstructured data returned by Twitter. These features were processed using feature selection and dimensionality reduction, and were then used to train popular machine learning models. Our framework could efficiently separate anti-government sentiment with approximately 87% accuracy, balancing precision and recall (F1-score of 0.82), and with a training time of only a few seconds. This anti-government propaganda was immersed in various contextual and circumstantial information, hence, lacked clear focus or philosophy. The research thus demonstrated the promise of feature engineering and machine learning to detect deviant content that can precipitate violence even though it is submerged in the surrounding events. It thus opens up the possibility of employing these techniques to identify and demote deviant chatter on social media platforms before it can cause offline damage, as well as gateways for future advances on this topic.

Our future research involves building methods to geo-locate tweets to understand the geographical dispersion of anti-government, extremist content.

### Funding
The authors received no funding for this work. The funders had no role in study design, data collection and analysis, decision to publish, or preparation of the manuscript.

### Competing Interests
The authors declare that they have no competing interests.

### Author Contributions
- Hieu Nguyen conceived and designed the experiments, performed the experiments, analyzed the data, prepared figures and/or tables, authored or reviewed drafts of the article, and approved the final draft.
- Swapna Gokhale conceived and designed the experiments, performed the experiments, analyzed the data, prepared figures and/or tables, authored or reviewed drafts of the article, and approved the final draft.

### Data Availability
The data is available at GitHub: https://github.com/HieuQN/Michigan-Protest.

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
