# Peer review of "An efficient approach to identifying anti-government sentiment on Twitter during Michigan protests"

_PeerJ Computer Science, doi:10.7717/peerj-cs.1127_

## Round 0.1 · original submission · Major Revisions

Three reports have been received. The comments regarding experimental design and statistics should be carefully addressed. Please provide a revised paper together with a detailed response letter. Thanks.

·

Basic reporting

Excellent command of English language was used along with great structuring. However, It was mentioned that (Wikipedia, 2020) was cited at lines 90, 92, 102 and 123, since it is not accredited scientific reference, it may be used once but not that much and at experimentation settings sections, so these has to be changed.

Experimental design

Experiments' settings and formulas along with findings and discussion were presented thoroughly which is great. Nevertheless, at line 208 expressions included are vague, their mathematical formula should be included first then to justify values and constants so please work on that.

Validity of the findings

No comments

Reviewer 2 ·

Basic reporting

The structure differs from the standard sections of PeerJ. It would be better and more desirable to reformat this paper according to PeerJ specifications.

The discussion section has been omitted. The section on related research that follows the section on experimental results seems to belong in the introduction. Also, its final paragraph is somewhat disconnected from the preceding paragraphs; a missing link is required to connect it to the literature presented in the preceding paragraphs.

Experimental design

By definition, neutral means not taking side, either with or against the government. However, some if not all examples of "Neutral" in Table 1 resemble "Pro-Government" rather than "Neutral." Also, as the number of tweets labeled "Neutral" in the dataset is roughly three times that of tweets labeled "Anti-Government", does this imply that the size of Pro-Government tweets is three times that of Anti-Government tweets during Michigan Protests? Because there may be genuinely neutral tweets in the mix, so the answer may be no.

On the other hand, why only focus on "Anti-Government" tweets when "Pro-Government" tweets can also be problematic? See:

Stukal, D., Sanovich, S., Bonneau, R., & Tucker, J. A. (2022). Why Botter: How Pro-Government Bots Fight Opposition in Russia. American Political Science Review, 1-15.

Caldarelli, G., De Nicola, R., Del Vigna, F., Petrocchi, M., & Saracco, F. (2020). The role of bot squads in the political propaganda on Twitter. Communications Physics, 3(1), 1-15.

Having said that, would it be better to classify the tweets into three, i.e., "Anti", "Neutral", and "Pro"? However, a great deal will depend on the research question, which is currently rather vague. Is this paper's proposed framework limited to tweets during the Michigan Protests? Will it apply to other events of a similar nature in the United States or the rest of the world?

Validity of the findings

What statistical tests are used in Table 3 and Table 4? Please report them appropriately.

Instead of merely mentioning them briefly after presenting them in Figure 1, it would be preferable to also present the results of text feature computation, particularly VADER, Readability, and Emotion, along with their summary statistics in the form of tables.

Reviewer 3 ·

Basic reporting

The research follows a more recent trend that studies radical extremism on social media platforms within the United States. It aims to identify anti-government tweets that try to erode trust in political institutions by spreading extreme, off mainstream content. The study surrounds two protests, “Operation Gridlock” and “Michigan Protest”, in Michigan. The background is not sufficient enough to know if the two protests are all the anti-lockdown protests in Michigan or they are the most representative examples. Overall, the paper is written with professional English with clarity. Minor typos include writing “through” as “though” and using present tense for some past events. The term “original tweet” is unclear to define if it is the tweet in the corpus or the first original tweet being quoted by subsequent tweets and followers. All figures and tables are relevant and well labelled except table 4 which is not indicated in the paragraphs. When describing figure 4, figure 1 is also mentioned, but “the figure” is used afterwards making it ambiguous about which one should be followed. The raw data seems also not supplied.

Experimental design

Traditional methods with designed feature engineering. The author shows improved efficiency by three levels of feature engineering based on statistical significance, use of important features and dimension reduction. This narrows down subsets of features to train popular machine learning classifiers. Both the structured and unstructured data have been utilized to facilitate the analysis of radical comments. DistilBERT model is used to score for some auxiliary features upon basic text features. However, Word2Vec model is used for the main text instead of transformer models.
On the other hand, structural data including interaction features and authors’ features are used as additional features. In interaction features, TF-IDF is calculated for the number of hashtags and mentions but not for the number of quotes. It should detail the difference in characteristics between them and why quotes are not matching with TD-IDF calculation. In authors’ features, when analyzing the data statistics of network strength, there is no conclusion on its significance in identifying the two groups anti-government (‘A’) or neutral (‘N’). Lastly, five themes are observed but only one is chosen for analysis. Is the annotation also categorized for the 5 themes or concluded from text analysis? It would be more transparent to present the statistics of the themes especially the proportion of text in Theme 1 that is related to the two protests. One of the contributions is to provide unique and early insights in chaotic circumstances brought about by the Covid-19 pandemic, but the theme for Covid-19 is not under analysis. It is concluded that social media feeds can be mined to provide unique and early insights into people’s radical opinions and thoughts “regardless of the context”, it will be interesting to learn some insights from the other themes.

Validity of the findings

In the abstract, it denotes the results with accuracy of 87%, which seems accuracy is the main evaluation in this work instead of F1-score. In word cloud visualizations, prominent words with high frequency are printed out, but the presentation of sample texts including those words would give a better derivation of the related concepts. For example, in text features, it is questionable why assholes, thugs, clowns, magats, cowards, mob and domestic terrorists are President-Trump-related. In authors’ features, it is asserted that no significant difference in the number of likes by the authors of tweets from both groups, but the p-value in Table 3 indicates the opposite. Last but not least, the labels categorize in two sides - anti-government & neutral. It would provide more distinct results by showing how to separate neutral stance from pro-government one.

---

## Round 0.2 · Minor Revisions

There are still some lingering comments from the reviewers. Please address them in the revised version. Thanks.

·

Basic reporting

No Comment

Experimental design

No Comment

Validity of the findings

No Comment

Additional comments

All comments have been taken into consideration as the new version shows

Reviewer 2 ·

Basic reporting

The authors restructured the manuscript, which I believe makes it more suitable for publication in PeerJ.

Experimental design

"... we chose to label the tweets into two groups, one group consisted of anti-government tweets, and the second group included tweets that are not against the government."

This sentence is clear enough, so I fail to understand why the authors switch back to using "anti-government" vs "neutral" as opposed to "anti-government" vs "not anti-government" later in the same paragraph, especially when they agreed that "the second group may contain a mix of pro-government and neutral tweets".

Validity of the findings

The authors clarified the use of two-sample t-tests in their results.
The authors also added the results of text feature computation, i.e., VADER, Readability, and Emotion in Table 3.

Reviewer 3 ·

Basic reporting

This is my 2nd time reviewing the paper. Therefore, I pay more attention to 1) the revised sections of the paper and 2) the raised concerns being addressed or not. The authors have addressed most of the concerns in the revision. However, some concerns should be further clarified before the paper is published.

Experimental design

The authors have addressed most of the comments except why Word2Vec model is used for the main text instead of transformer models or other pretrained models. Including pretrained models as the comparison will make this paper more complete, since the pretrained model is the mainstream method in the NLP research field.

Validity of the findings

1. The authors have mentioned that only for the first theme examples from both Operation Gridlock and Michigan Protest data sets are included. Do they exclude examples of the protests in other themes? How do they analyze if the examples pertain to both datasets? Moreover, they did not identify the specific theme associated with the tweet, and not feasible to provide the split of the tweets into these five themes. Then, how did they choose the first theme examples from both Operation Gridlock and Michigan Protest data sets? It is necessary to separate the first theme from the others in order to do so.

2. The authors claimed that the president is blamed for Covid deaths and the anti-government trend was amplified during the Covid-19 pandemic. However, the results from word cloud do not show a direct indication of Covid-19.

---

## Round 0.3 · accepted · Accept

The authors have addressed all reviewers' concerns. I recommend it for publication.